# Effects of different tennis racket string tension on forehand stroke effect and racket dynamic impact

**Guohui Zhao**[1,2], **Chen Li**[1☯], **Ye Liu**[1☯]*

**1** School of Sport Science, Beijing Sport University, Beijing, China, **2** School of Public physical Education, Shanxi University, Taiyuan, China

☯ These authors contributed equally to this work.
* liuye_tg@163.com

**Data Availability Statement:** All relevant data are within the manuscript and its Supporting Information files.

**Funding:** The author(s) received no specific funding for this work.

## Abstract

This study investigates the effect of varying tennis racket string tension on stroke effect and the dynamic response of the racket. Using the YSV dynamic acceleration signal acquisition system and a portable radar speed gun collect data on racket acceleration, stress-strain signals, and ball speed from 15 male athletes. Stroke accuracy and depth were assessed according to the International Tennis Number. The recorded stroke speeds were 108.87 ±13.57 km/h, 111.83±16.34 km/h, and 107.76±12.53 km/h for the low, medium, and high tension, respectively. A significantly higher ball speed was observed at 54lbs compared to both 48lbs and 60lbs (P <0.05). Control scores were 4.90±0.61,5.46±0.84, and 4.64±0.69 for the respective tensions. The control at 54lbs was significantly higher than at 60lbs (P<0.05). Deformation measurements were 18.53±4.90µε, 16.31±4.42µε, and 20.90 ±3.53µε, with significantly lower deformation at 54lbs compared to 60lbs (P<0.05). The impact forces recorded were 381.81±48.51m/s$^2$, 380.53±50.47m/s$^2$, and 380.04±53.70m/s$^2$, with no significant effect of string tension on impact force. Racket vibration frequencies were 44.14±0.48Hz, 44.08±0.35Hz, and 44.14±0.25Hz, with no significant difference among three string tensions. Rackets with three string tensions showed significantly higher vibration frequencies during the collision phase compared to before or after (P<0.01). In conclusion, string tension affect the stroke effect, racket strung at medium tension can optimize stroke effect while got milder dynamic impact, suggesting that racket strung at medium tension is recommended for tennis enthusiasts to enhance stroke performance and to decrease the risk of resonance damage in the forearm soft tissues.

## Introduction

The forehand stroke, ranking as the second most vital stroke in tennis [1], is the most commonly executed groundstroke among professional players [2, 3] and typically the initial skill acquired by novice tennis enthusiasts [4]. Mastery of the forehand is essential for maintaining an aggressive playing stance [5]. This stroke constitutes over 25% of all tennis actions and is

**Competing interests:** The authors have declared that no competing interests exist.

recognized as a significant indicator of match outcomes [6, 7]. Ball velocity is a paramount determinant of tennis performance, as it limits the opponent's time to prepare and execute their response [7]. Additionally, ball precision is a critical performance metric in tennis [8], and when combined with optimal ball velocity, it can dictate the success of a rally [7]. The velocity and precision of the ball are the principal determinants of stroke excellence [9].

The tennis racket consists of the frame, handle, and strings [10]. The International Tennis Federation permits manufacturers to modify the racket's stiffness, weight, balance, and string pattern, which in turn influence the racket's playing characteristics. Among these factors, string tension plays a crucial role in determining both the playing performance and material characteristics of the racket. Evidence suggests that both professional and amateur tennis players change their string tension before matches or during training in order to minimize the impact of changes in string tension on their shot performance [11].

String tension impacts stroke effectiveness, with racket speed and stroke accuracy being critical determinants of tennis performance [8]. It is commonly believed that within the recommended range of string tension, lower tension results in greater rebound speed [12–16]. However, rackets with lower tension did not produce higher stroke speeds [11]. Conversely, higher tension enhanced ball control, with no significant speed differences between medium and low-tension strings [17]. They also reported varied success rates in strokes across different tensions, with no consistent outcomes. Notably, rackets with a medium tension of 53-pound (235N) showed the highest baseline errors, suggesting no significant correlation between control and string tension [18]. Previous investigations predominantly assessed the effects of string tension on ball velocity within controlled laboratory settings [14, 19–26], where criteria for evaluating control were frequently arbitrary, devoid of objective benchmarks that reflect the players' actual performance in stroke control [11, 17, 26]. In stark contrast, the International Tennis Number (ITN) rating test stands as a globally acknowledged benchmark for tennis proficiency, delineating precise guidelines for the demarcation of tennis courts and the assessment of stroke control.

String tension also influences the racket's dynamic response [27]. Research indicates that higher string tension generate more impact vibrations during strokes. The vibration frequency in the holding hand of participants maintaining a stroke posture ranged from 0–120 Hz when the racket was struck at a certain speed [28]. Through analyzing racket frame acceleration during simulated matches, found the vibration frequency of the racket frame to be around 200 Hz [29]. But the racket's inherent vibration frequency lies between 130–160 Hz [21]. These studies suggest that changes in string tension induce corresponding vibrations in the racket [27], yet they predominantly focused on comparing vibrations of hand-held, freely suspended, or clamped rackets [26, 28–34], neglecting the real-life context of tennis strokes and human participation. They did not analyze the racket's dynamic response during athletes swinging and stroking processes.

Previous studies on the effects of string tension on stroke effect lack quantitative evaluations, and overlook the human element in real tennis strokes, thus detaching the studies from actual play conditions. Given the significance of forehand strokes [35], this investigation focuses on forehand strokes in conjunction with the International Tennis Number(ITN). The study hypothesizes that with increasing string tension, stroke control improves, stroke speed decreases, and racket vibrations increase. The study aims to analyze the impact of string tension variations on stroke control and the characteristics of racket vibrations during strokes. The goal is to provide tennis enthusiasts with more informed recommendations on string tension selection.

**Table 1. The participants information.**

| Number of Participants | Gender | Age (years) | Height (cm) | Weight (kg) | BMI (kg/m²) | Expertise (years) | Training per Week (hours) |
|---|---|---|---|---|---|---|---|
| 15 | Male | 24.20±0.74 | 179.00±5.90 | 71.00±8.58 | 22.15±2.17 | 5.36±0.84 | 29.50±7.77 |

## Methods

### Participants

Fifteen male students specializing in tennis at Beijing Sport University, all of whom are right-handed, were recruited for this study. None of the participants had a history of injury within the six months preceding the tests, and none engaged in intense physical activity during the week before the tests. After providing written informed consent, their information was recorded and is presented in Table 1. All procedures were conducted in conformity with the Declaration of Helsinki and were approved by the Ethical Review Board of Beijing Sport University Sports Science Experiment Ethics Committee (2021060H).

### Experimental procedures

Before the formal tests, participants were familiarized with the testing procedures under standardized guidance. Formal Testing: Participants stood at the baseline of a standard tennis court (23.77meters long and 8.23meters wide) and, using their preferred stance, hit the ball with maximum effort using a forehand stroke directed cross-court, each to each performed three sets of tests, with a 15-minute rest period between each set. Before each set, participant performed two practice strokes, followed by six test strokes. All forehand stroke were executed using a semi-Western grip with maximum force. Racket acceleration, stress-strain signals, ball speed, and scoring were recorded.

The tennis court was divided into specific zones (Fig 1) according to the International Tennis Number (ITN), and ball landing spots were recorded based on ITN. Throughout the experiment, every participant used rackets with string tension of 48lbs, 54lbs, and 60lbs [11, 22, 29, 36] in a randomized order for the three sets, coach delivered the balls the in sync with the participants hitting rhythm though self-made, simple ball-feeding device(Fig 2), positioned as shown in Fig 1. Ball speed was measured using a portable radar speed gun that's supported by a triangular bracket at 3.50meters from baseline (Stalker Pro II, Stalker Radar, Plano, Texas, USA). An 8-channel YSV8008 acceleration acquisition system (Beijing Yiyang Strain and Vibration Testing Technology Co., Ltd., China, sampling rate: 2048 Hz) and a YSV2303S accelerometer (Beijing Yiyang Strain and Vibration Testing Technology Co., Ltd., range: ±100g, sampling rate: 2048 Hz, weight: 18g) recorded the racket's impact during strokes. Racket deformation was measured using BX120-2AA stress-strain gauge. The placement of the accelerometer and stress-strain gauge is shown in Fig 3. The tests utilized tennis racket (Wilson k tour 95), TELOON tennis balls, and racket strings with a diameter of 1.15 mm polyester. The portable radar speed gun was positioned 3.50 meters from the baseline and 3.30 meters from the singles sideline (Fig 1). The self-made ball-feeding device was placed 2.29 meters from the center of the device and 2.30 meters from the singles sideline, as indicated in Fig 1.

### Data processing

In this study, an accelerometer was used to collect the variation of acceleration along the Z-axis, perpendicular to the racket face, during forehand strokes. This measurement reflects the maximum impact force exerted on the racket in the vertical direction during the stroke. All

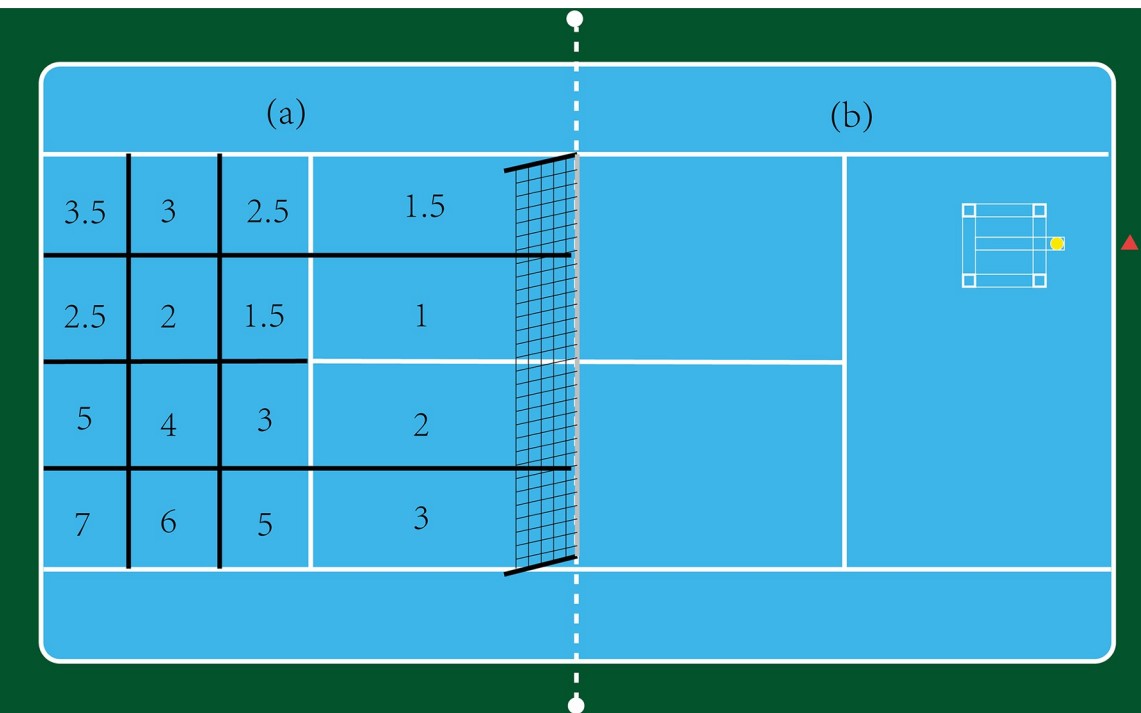

**Fig 1. Tennis court division and ball feeder location.** (a) According to the International Tennis Number (ITN) grading system tennis court was divided into specific zones, (b)The placement of ball feeder, The triangle represents the position of portable radar speed gun.

experimental data processing was based on valid strokes. The stroke control score was derived by referencing the depth and accuracy scores from the International Tennis Number (ITN).

The YSV data acquisition system applied a 20–220 Hz bandpass Butterworth filter to the data recorded by the accelerometer [37]. The vibration frequency of the racket before, during, and after the stroke, as well as the maximum strain signal during the stroke, were calculated. The average values of the data from each participant's valid strokes were used as the final data. Data processing and statistical analysis were performed using custom-written MATLAB scripts (The MathWorks Inc., USA) and SPSS 17.0 (SPSS Inc., USA). The Pauta criterion was used to exclude outliers from the data set if their deviation from the mean exceeded twice the standard deviation. After removing these outliers, paired T-tests were conducted. Pearson correlation analysis was employed to examine the relationships among the data. The results were presented as mean ± standard deviation (M±S). Statistical significance was defined as a type I error probability not exceeding 0.05.

## Results

### Impact of different string tension on the effect of stroke

In terms of ball speed, the racket strung at 54lbs produced the highest speed, followed by the racket strung at 48lbs, with the racket strung at 60lbs producing the lowest speed. There was no significant difference in ball speed between the 48lbs and 54lbs rackets (P = 0.036). Similarly, there was no significant difference in ball speed between the 48lbs and 60lbs rackets (P = 0.977). However, there was a significant difference in ball speed between the 54lbs and 60lbs rackets (P = 0.014). Thus, the highest ball speed was achieved with the 54lbs racket, followed by the 48lbs racket, and the lowest ball speed was achieved with the 60lbs racket.

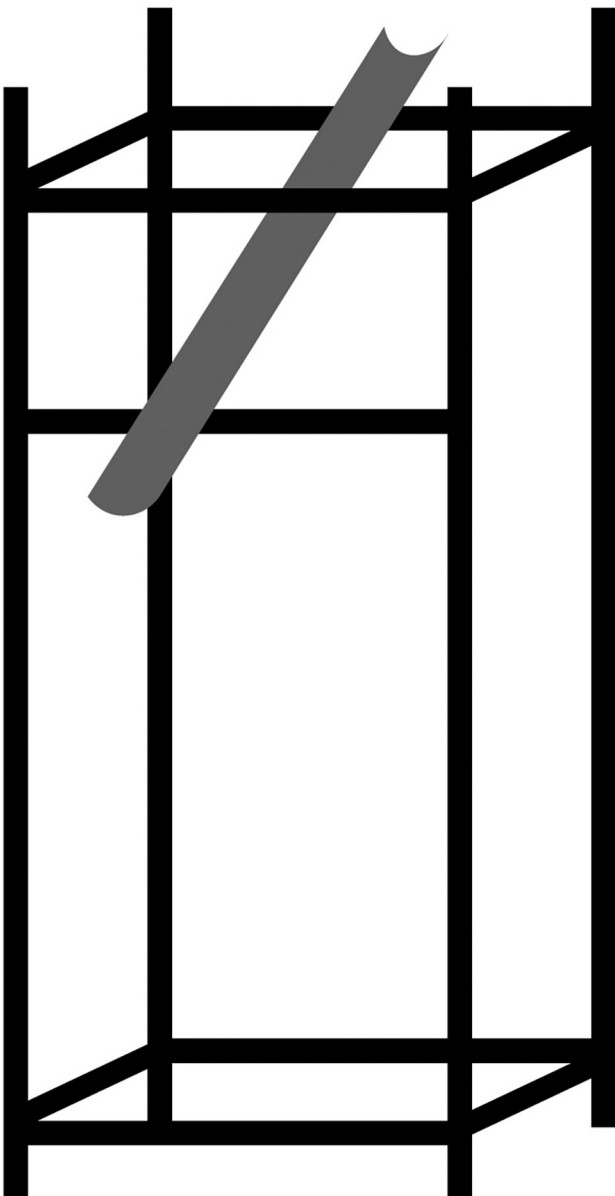

**Fig 2. Simple ball-feeding device.** The center of the self-made ball feeding device is located 229cm from the baseline and 230cm from the singles sideline. The height of the ball entrance end from the ground is 246cm, while the height of the ball exit end from the ground is 194cm. The ball exit end is 111cm from where the ball lands.

Regarding the score of control, there was no significant difference between the 48lbs and 54lbs string tension (P = 0.161), nor between the 48lbs and 60lbs string tension (P = 0.073). However, there was a significant difference in control scores between the 54lbs and 60lbs string tension (P = 0.012). The analysis indicates that different string tension can impact stroke control, with the 54lbs string tension providing the best control score, which is higher than both the 48lbs and 60lbs tension.

## The influence of different string tension on racket impact force

When the string tension is 48lbs, the resulting peak acceleration does not significantly differ from the peak accelerations produced at string tension of 54lbs and 60lbs (P = 0.468,

front

back

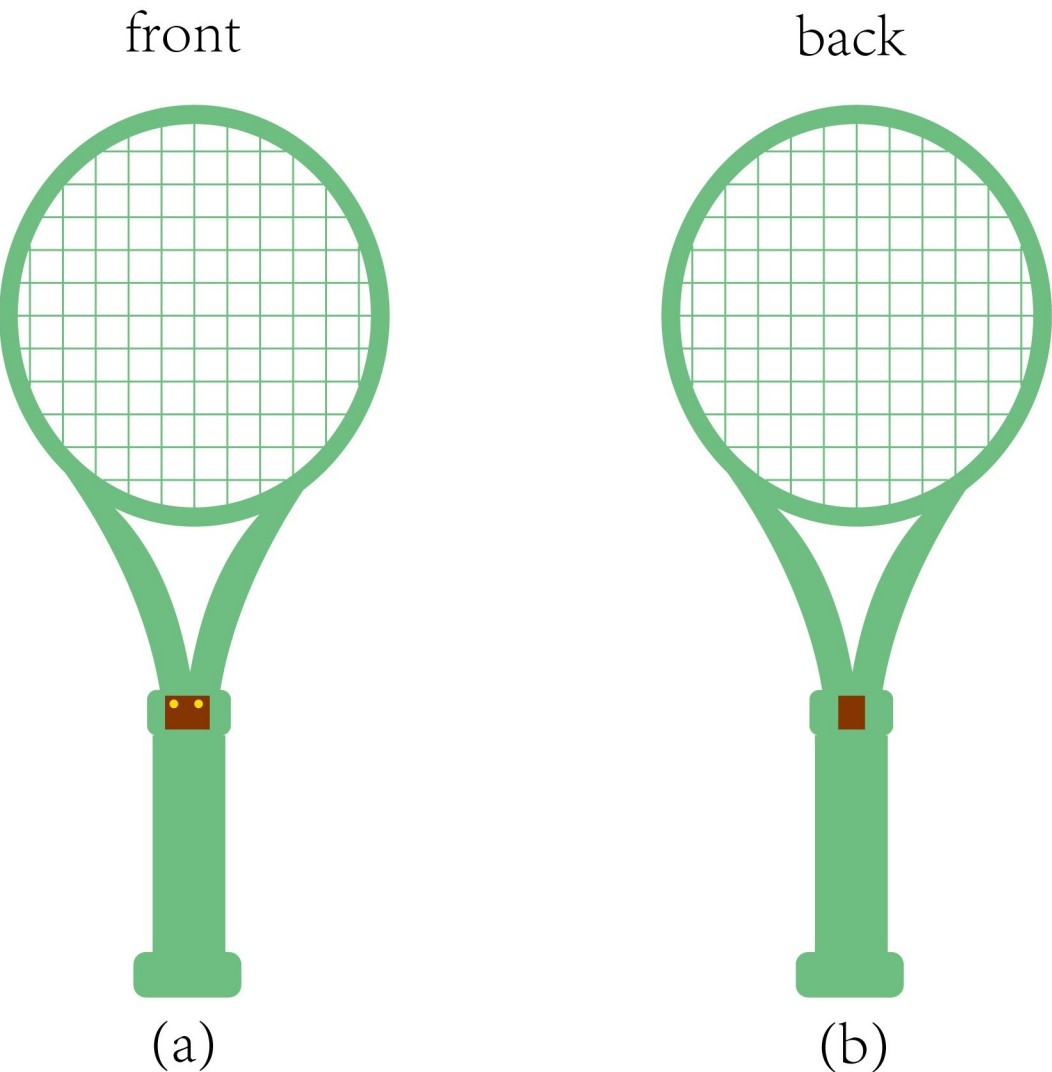

(a)

(b)

**Fig 3. The placement of the accelerometer and stress-strain gauge.** (a) The placement of the accelerometer, (b) The placement of stress-strain gauge.

P = 0.451). Similarly, there is no significant difference in the collision acceleration during impact between rackets strung at 54lbs and those strung at 60lbs (P = 0.390). The deformation of the racket at 48lbs during ball impact does not significantly differ from the deformation at 54lbs and 60lbs (P = 0.103, P = 0.908). However, there is a significant difference in the amount of deformation between string tension of 54lbs and 60lbs (P = 0.018). The string tension results in the racket experiencing essentially the same impact load, yet the deformation varies with different string tension. Specifically, the deformation is minimal at 54lbs, followed by 48lbs, with the greatest deformation occurring at 60lbs.

### Effect of different string tension on the vibration frequency of racket

Before the collision, the vibration frequencies generated by rackets with different string tensions showed no significant differences (P > 0.05). However, during the collision, the vibration frequencies increased significantly across all tensions levels due to the impact (P < 0.01), with

no significant differences between the different tensions groups (P > 0.05). In the follow-through phase after the collision, the vibration frequencies of rackets with different tensions rapidly decayed back to the pre-collision levels, again with no significant differences between the tension groups (P > 0.05).

## Discussion

By studying the impact of string tension variations on stroke effect and the dynamic response of the racket, the following key findings were observed: string tension significantly affects both ball speed and control. The 54lbs string tension resulted in the highest ball speed and best control, followed by 48lbs and 60lbs, string tension has minimal effect on the racket's dynamic impact force and vibration. However, it significantly influences racket deformation, with the greatest deformation occurring at 60lbs, followed by 48lbs, the least deformation at 54lbs.

### Effect of different string tension on stroke effect

The tension of tennis racket strings significantly impacts ball speed. The ball speed generated with a 54lbs string tension is greater than that with a 60lbs tension, and this difference is statistically significant (Table 2). This conclusion aligns with the findings that lower string tension results in faster ball speeds [11, 17, 38]. From an energy perspective, when the ball collides with the racket strings, kinetic energy is converted into elastic energy, which is evenly distributed between the racket and the ball [39, 40]. The elastic energy stored in the strings accounts for approximately 95% of the total elastic energy [41], while the energy stored in the tennis ball itself accounts for 5% [42]. The ball utilizes 55% of its own stored elastic energy [41], with more elastic energy being stored in the strings. Consequently, changes in energy distribution after the collision have a more significant impact.

Consistent with the results of this study, lower string tension leads to greater deformation of the strings and smaller deformation of the ball, resulting in lower overall energy consumption by the ball [13]. When the ball collides with higher string tension, it undergoes greater deformation, resulting in higher energy consumption [21, 24]. Since the total energy remains constant during the collision, greater string deformation corresponds to higher energy storage and more energy transferred back to the ball, resulting in faster ball speed. Therefore, lower string tension produces faster ball speeds compared to higher string tension. The rebound coefficient, representing the ratio of the velocity after collision to the velocity before collision between the ball and the racket face, is highest for rackets with lower string tension [16]. The increased rebound coefficient associated with lower tension is believed to be due to the trampoline effect of the racket. Lower string tension leads to a greater trampoline effect, causing more significant deformation of the racket string bed and storing more energy, thus increasing the energy transferred to the ball and explaining the reason for higher ball speeds at lower tension [43]. To substantiate this perspective, the study findings that demonstrate the most significant reduction in ball speed ratio. (vector sum of the pre-collision and post-collision ball

**Table 2. Influence of different string tension on ball speed.**

| Group | N | Ball speed (km/h) | Score of control |
|---|---|---|---|
| 48bl | 46 | 108.87±13.57Δ | 4.90±0.61 |
| 54bl | 41 | 111.83±16.34 | 5.46±0.84 |
| 60bl | 49 | 107.76±12.53Δ | 4.64±0.69Δ |

Note: "Δ" indicates a significant difference (P < 0.05) to stroke the ball with a string tension of 54lbs.

speeds divided by the pre-collision speed) occurs at a string tension of 250N (approximately 56lbs), with a decrease observed when the tension is lower or higher than 250N [13]. The relationship between ball speed and string tension in this study further confirms a sinusoidal function.

In terms of ball control, string with a moderate tension of 54lbs provide better control during ball impact compared to a higher tension of 60lbs, and this difference is statistically significant. The results of this study are consistent with the findings that moderate string tension result in better success rates for racket-ball impact [17], in contrast to the research [11, 44]. Laboratory studies have also indicated that higher string tension improves ball control, while lower string tension allows for higher ball speeds [14]. However, this relationship is effective only when the tension is approximately 40lbs or higher [22]. Lower string tension allows for faster ball speeds [13], whereas higher string tension favors ball control [17]. The results of this study suggest that string tension between low and high levels provide better ball control during impact.

## Dynamic impact of different string tension on tennis rackets

The effect of different string tension has been extensively investigated. Varied string tension can alter the stiffness of the racket materials, leading to different deformations when subjected to forces. The impact force experienced by the racket during a stroke is generated by the ball collision with the strings [22, 45]. The deformation of the racket results from the collision between the ball and the string bed. Acceleration data of the racket is related to the initial impact, and the maximum load, as well as tissue stress and strain, are influenced by the initial impact load [46]. However, the variation in string tension does not significantly affect the magnitude of the impact force on the racket face during low, medium, or high string tension strokes, with an average peak force of approximately 380N. This phenomenon is due to the increase in the racket trampoline effect caused by lower poundage, resulting in greater deformation of the racket face [43]. This theory can explain why the racket deformation differs despite the same applied force on the string bed in this study. Compared to the two relatively lower tension strings, the 60-pound tension string exhibits a lower trampoline effect and lesser cushioning, leading to larger deformation of the racket itself (Table 3). On the other hand, both medium and low string tension can generate a strong trampoline effect, resulting in similar deformations of the racket. The analysis suggests that different string tension lead to variations in the deformation of the racket caused by the collision between the ball and the strings. Higher string tension can cause greater deformation of the racket, potentially impacting control over the ball. This is supported by the findings of this study, where the 60-pound group demonstrated lower stroke control scores compared to the 48-pound and 54-pound groups, while the 54-pound group with the smallest racket deformation showed the highest stroke control scores (Table 2).

**Table 3. Influence of different string tension on peak acceleration in the vertical racquet face direction.**

| Group | N | Peak acceleration (m/s$^2$) | Racket strain (με) |
|---|---|---|---|
| 48bl | 46 | 381.81±48.51 | 18.53±4.90 |
| 54bl | 41 | 380.53±50.47 | 16.31±4.42 |
| 60bl | 49 | 380.04±53.70 | 20.90±3.53Δ |

Note: "Δ" indicates a significant difference (P < 0.05) compared to the deformation parameter of the racket when stroke the ball with a string tension of 54lbs.

**Table 4. Influence of different string tension on vibration frequency of the racket during stroke.**

| Group | N | Vibration frequency (Hz) | | |
|---|---|---|---|---|
| | | Before collision | During collision | After collision |
| 48bl | 46 | 39.76±1.23 | 44.14±0.48ΔΔ | 39.45±0.96## |
| 54bl | 41 | 40.59±1.30 | 44.08±0.35ΔΔ | 39.54±1.36## |
| 60bl | 49 | 39.77±0.81 | 44.14±0.25ΔΔ | 39.05±1.08## |

Note: "ΔΔ" indicates a significant difference ($P < 0.01$) compared to the racket vibration frequency before the collision, "##" indicates a significant difference ($P < 0.01$) compared to the racket vibration frequency during the collision.

In addition to the materials and structural characteristics of the racket itself, the influence of hand grip cannot be ignored in the dynamic performance of the racket [47]. During a stroke, the interaction between the racket and the arm alters the racket dynamic performance, as it experiences the impact of the ball colliding with the strings as well as the racket vibration [48]. The vertical vibration of the racket face reflects its performance [37], with the majority of vibration frequencies ranging from 80 to 300 Hz. Throughout the entire stroke process with three different string tension, the vibration frequency is highest during collision but still lower compared to static and simulated tests, averaging only around 44 Hz (Table 4). Moreover, the vibration frequency of the racket quickly decays to pre-collision levels after the stroke, as the vibration energy transfers to the forearm [48], triggering active contraction of the forearm muscles [49] and changes in the dynamic viscoelasticity of the forearm soft tissues [48]. These variations in the forearm soft tissues help avoid resonance damage by shifting the vibration frequency away from the racket vibration frequency during the stroke.

This study investigates the effects of string tension on both stroke efficacy and the dynamic behavior of tennis rackets under actual playing conditions. Within the acceptable tension parameters, 54-pound strings were found to yield greater ball velocity and enhanced control during play, with negligible impact on the racket dynamic performance. The soft tissues of the upper limb actively respond, mitigating resonance and consequently diminishing the likelihood of forearm injuries. This is particularly evident as the racket undergoes minimal deformation upon ball impact when the force of impact is relatively constant. Thus, for players and coaches, selecting a medium string tension within the racket permissible range is beneficial not only for improving stroke effect but also for minimizing the risk of arm injuries due to vibrations and impact forces during play. For racket manufacturers, it is recommended to optimize the racket's trampoline effect to absorb the impact load during the stroke, which can reduce the incidence of forearm injuries and the development of tennis elbow.

This study has several limitations, including the omission of grip strength and stroke position measurements, both of which are known to affect racket vibrations and impact forces. To address this, we utilized a custom-designed ball-feeding device to standardize ball speed and simulate realistic stroke scenarios. Intermediate-level players were involved to minimize variability associated with these factors. Data analysis was performed using paired sample comparisons to minimize biases arising from individual differences. The scope of the research was confined to a single racket brand and three specific string tension. Future research is intended to include a variety of brands and a broader spectrum of string tension.

## Conclusions

From the perspective of the impact on the racket-arm system, medium string tension generates the highest ball speed and best control during play. Therefore, it is recommended that tennis

enthusiasts opt for rackets with medium string tension for optimal stroke performance. Additionally, racket strung at medium tension got milder dynamic impact, suggesting that manufacturers improve the racket's trampoline effect and provide cushioning to mitigate the loads produced during collisions.

## Supporting information

**S1 File. Raw data.**
(ZIP)

## Acknowledgments

The authors would like to acknowledge the valuable contribution of all the athletes and researchers who participated in this study.

## Author Contributions

**Conceptualization:** Chen Li.

**Data curation:** Chen Li.

**Methodology:** Chen Li.

**Project administration:** Ye Liu.

**Supervision:** Ye Liu.

**Writing – original draft:** Guohui Zhao.

**Writing – review & editing:** Guohui Zhao.

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
