## [Decision Letter · Decision Letter 0]

7 Feb 2024

PONE-D-24-00255Effects of different tennis racket string tension on forehand stroke effect and racket dynamic impactPLOS ONE

Dear Dr. Liu,

Thank you for submitting your manuscript to PLOS ONE. After careful consideration, we feel that it has merit but does not fully meet PLOS ONE’s publication criteria as it currently stands. Therefore, we invite you to submit a revised version of the manuscript that addresses the points raised during the review process.

We look forward to receiving your revised manuscript.

Kind regards,

Yaodong Gu

Academic Editor

PLOS ONE

Journal Requirements:

2. We note that your Data Availability Statement is currently as follows: "All relevant data are within the manuscript and its Supporting Information files"

4. Please upload a new copy of Figures 1 and 2 as the detail is not clear. Please follow the link for more information: 

https://blogs.plos.org/plos/2019/06/looking-good-tips-for-creating-your-plos-figures-graphics/

https://blogs.plos.org/plos/2019/06/looking-good-tips-for-creating-your-plos-figures-graphics/

5. Please remove your figures from within your manuscript file, leaving only the individual TIFF/EPS image files, uploaded separately. These will be automatically included in the reviewers’ PDF.

6. We note you have included a table to which you do not refer in the text of your manuscript. Please ensure that you refer to Tables 2 and 3 in your text; if accepted, production will need this reference to link the reader to the Table.

**Additional Editor Comments:**

Please put more information in the methods part as reviewer request.

Reviewers' comments:

Reviewer's Responses to Questions

**Comments to the Author**

1. Is the manuscript technically sound, and do the data support the conclusions?

Reviewer #1: Partly

Reviewer #2: Yes

2. Has the statistical analysis been performed appropriately and rigorously? 

Reviewer #1: Yes

Reviewer #2: Yes

3. Have the authors made all data underlying the findings in their manuscript fully available?

Reviewer #1: Yes

Reviewer #2: Yes

4. Is the manuscript presented in an intelligible fashion and written in standard English?

Reviewer #1: Yes

Reviewer #2: Yes

5. Review Comments to the Author

Reviewer #1: Dear Author/s,

the Author/s did a good job in writing a readable manuscript. This paper could be considered qualified to be published on Plos One if the authors apply the modifications requested, "Major issues" are recommended. It is the reason why "Major revision" is my personal choice. I would be happy to review a revised version of this manuscript.

The aim of the submitted manuscript was to control explore the impact of changes in tennis racket string tension on hitting performance and racket dynamic response.

This paper could be considered qualified to be published if the authors apply the modifications requested.

While it is a very interesting topic, some suggestions for improving the paper are provided below:

• The basic descriptive level adopted by author(s) does not seem compatible with interests of both researchers' conceptual challenges and coaches' practical applications to training. The main concern refers to the weak/absent theoretical background; there is no perceivable research question or rationale for the study presented to readers.

Author(s) do not consider any specific difficulties or other aspects of performing these different types of tennis techniques to minimally justify the comparison; on the other hand, the use of the descriptive knowledge obtained from the observed differences was not discussed in terms of applications for athletes and coaches. In short, the manuscript is limited to a laboratory exercise to partially describe the participants' movement.

• Introduction. Line 71: the Aim of the study has to be included at the end of the Introduction section.

• Subject and Methods. Line 74: more detailed information about participants have to be included in this section (mean values and standard deviation): age, weight, eight, handedness, expertise (years), hours of training per week, etc.

• Line 86. A portable radar speed gun was employed. Details about it have to be reported in the manuscript: brand, model, etc.

• Figures 1 and 2 don’t reach a minimum standard of quality. The two pictures need to be removed from the paper and replaced by high quality pictures or figures/drawings.

• Line 83-85: there are many typing errors.

• Line 95: a feeding machine was used in the present investigation. The brand needs to be specified.

• The balls were dropped by a feeding machine in a specific area of the tennis court. This area has to be described by Authors.

• Line 97: information about target area (dimension and position) can not totally excluded from the paper. I would suggest to include a figure about the set-up.

• Line 102: the subjects randomly used 48-54-60 lbs string tension. Authors have to justify this decision and support it by references.

• Line 102. There are errors about unite of measure.

• Line 126: more details about “YSV data acquisition system” have to be included in the present section.

• Line 139-139. The title of the sub-section needs to be modified. Indeed, “stroke effect” seems not to describe the date present in the Table 1.

• Discussion: a Limits of the study section needs to be included in the present investigation.

• Discussion: some practical applications have to be reported at the end of the manuscript.

• Line 261-262. This sentence has to be removed from the Conclusion and included in the Methods.

Finally, this manuscript could be considered qualified to be published if the authors apply the previous modifications, therefore “Major revision” is recommended.

Reviewer #2: Thank you for the opportunity to review this unique study outlining tennis racket string tension. Overall, this manuscript is well-written, stating that the paper is interesting and correctly prepared.

The following are editing suggestions:

1.In the abstract part, methods, it is essential to add the something about statistical analyses.

2.In the abstract part, results, main findings are needed to show and combine in this part.

3.In the intro part, the first paragraph, “The tennis racket is a complex system composed of the frame, handle, and strings” It is recommended that add some relevant references here.

4.In the intro part, display the aim of this study rather than the necessary and hypothesis.

5.In the methold part, the method is well-structured, with a clear and correct development of sections. It is better to add a figure of experimental set-up which includes participant, racket, net and other necessary equipment.

6.In the results part, tables included in the paper are clear, and there is extensive notes accompanying them. This suggests that the visual representations are effective in conveying information. However, the content of the results text is too short, try to add some details in order this section entirely.

7.In the discussion part, the authors should display the limitations of their study,which is appropriately discussed in the end.

6. PLOS authors have the option to publish the peer review history of their article (what does this mean?). If published, this will include your full peer review and any attached files.

Reviewer #1: No

Reviewer #2: No

---

## [Author Response · Author response to Decision Letter 0]

1 May 2024

Responds to the reviewers’ comments:

Reviewer #1:

General comments：

Comment 1: the Author/s did a good job in writing a readable manuscript. This paper could be considered qualified to be published on Plos One if the authors apply the modifications requested, "Major issues" are recommended. It is the reason why "Major revision" is my personal choice. I would be happy to review a revised version of this manuscript.The aim of the submitted manuscript was to control explore the impact of changes in tennis racket string tension on hitting performance and racket dynamic response.This paper could be considered qualified to be published if the authors apply the modifications requested.

Response 1: We appreciate the reviewer for the favorable analysis of our original submission and for highlighting the significance of our studies.

Comment 2: The basic descriptive level adopted by author(s) does not seem compatible with interests of both researchers' conceptual challenges and coaches' practical applications to training. The main concern refers to the weak/absent theoretical background; there is no perceivable research question or rationale for the study presented to readers. Author(s) do not consider any specific difficulties or other aspects of performing these different types of tennis techniques to minimally justify the comparison; on the other hand, the use of the descriptive knowledge obtained from the observed differences was not discussed in terms of applications for athletes and coaches. In short, the manuscript is limited to a laboratory exercise to partially describe the participants' movement.

Response 2: Thank you for your review comments. We have revised the introduction according to your suggestions. The first paragraph of the introduction now presents the research background, emphasizing the significance of string tension in impacting the effectiveness of the stroke and the dynamic response of the racket. The second and third paragraphs discuss the current research on the effects of string tension on the stroke outcome and the state of research on racket dynamics in terms of impact and vibration, as well as the shortcomings in these areas. The fourth paragraph summarizes these issues, noting the lack of quantitative control and evaluation standards in terms of stroke effectiveness, and the oversight of human involvement in practical situations in the research on impact and vibration.

In tennis, the forehand stroke is second only to the serve in importance, and forehand techniques can be divided into topspin and flat strokes. The flat stroke, being an important scoring method that requires high control precision, was the focus during our testing process. We asked all participants to use the forehand flat stroke technique to complete the tests, which eliminated the variability that different tennis techniques might introduce. specific difficulties and other aspects of performing these different types of tennis techniques have been discussion in line 300-309 of manuscript.

The article has been expanded to include discussions about the use of the descriptive knowledge obtained from the observed differences in terms of applications for athletes and coaches in line 285-296 of manuscript .

Specific comments:

Comment 1: Introduction. Line 71: the Aim of the study has to be included at the end of the Introduction section.

Response 1: The aim of the study has been included in the line 71-74 of manuscript.

Comment 2: Subject and Methods. Line 74: more detailed information about participants have to be included in this section (mean values and standard deviation): age, weight, eight,handedness, expertise (years), hours of training per week, etc.

Response 2: We are sorry for our unclear descriptions of the Participants , Participants detailed information are included in Table1 in the line 83 of manuscript

Comment 3: Line 86. A portable radar speed gun was employed. Details about it have to be reported in the manuscript: brand, model, etc.

Response 3: We are sorry for our unclear descriptions of the portable radar speed gun ,Details about the radar speed gun has been reported in the line 95-96 of manuscript 

Comment 4: Figures 1 and 2 don’t reach a minimum standard of quality. The two pictures need to be removed from the paper and replaced by high quality pictures or figures/drawings.

Response4: According to the requirements of the journal picture, the quality of the picture has been modified. See the attachment for the modified picture.

Comment5: Line 83-85: there are many typing errors.

Response 5: We are sorry for typing errors. The typing errors has been corrected in lines85-87of manuscript

Comment 6: Line 95: a feeding machine was used in the present investigation. The brand needs to be specified.

Response 6: We are sorry for our unclear descriptions of feeding machine ,In order to ensure that the subjects hit the ball with the maximum power of forehand, we used a self-made simple ball feeding device, and adjusted the device to make the ball in the most suitable hitting position, which can effectively avoid the instability of the machine feeding ball. The picture of the device is illustrated in the Fig3.

Comment 7: The balls were dropped by a feeding machine in a specific area of the tennis court. This area has to be described by Authors.

Response 7: We are sorry for our unclear descriptions of where is feeding machine, the specific area of the tennis court is described in the line98-101, which is shown by the triangle in Fig4

Comment 8: Line 97: information about target area (dimension and position) can not totally excluded from the paper. I would suggest to include a figure about the set-up.

Response 8: We are sorry for our unclear descriptions of target area, information about target area is shown in the Fig4

Comment 9: • Line 102: the subjects randomly used 48-54-60 lbs string tension. Authors have to justify this decision and support it by references.

Response 9: The racquet used for testing has a recommended racquet tension of 40-60 pounds, combined with previous studies（Fabre, 2014;Zhao, 2010; Bower, 2007;R.Cross, 2001）. So the 48-54-60 lbs string tension was selected

Comment 10: • Line 102. There are errors about unite of measure.

Response 10: Errors about unite has been corrected in lines 113 of manuscript 

Comment 11: Line 126: more details about “YSV data acquisition system” have to be included in the present section.

Response 11: The details about “YSV data acquisition system” included in the line 86-95 of manuscript

Comment 12: Line 139-139. The title of the sub-section needs to be modified. Indeed, “stroke effect” seems not to describe the date present in the Table 1.

Response 12: In the line 139-140, the title of the sub-section was modified to “Impact of different string tensions on the effectiveness of stroke”

Comment 13: Discussion: a Limits of the study section needs to be included in the present investigation.

Response 13: We apologize for the inappropriate discussion here, the Limits of the study was included in the line 301-310 of manuscript 

Comment 14: Discussion: some practical applications have to be reported at the end of the manuscript

Response 14: We apologize for the inappropriate discussion here, the practical applications have been reported in line 288-300of manuscript

Comment 15: Line 261-262. This sentence has to be removed from the Conclusion and included in the Methods.

Response 15: Your suggestion is quite reasonable . The sentence in line 261-262 has moved from the Conclusion to line 101-102 in the Methods of manuscript 

Reviewer #2: Thank you for the opportunity to review this unique study outlining tennis racket string tension. Overall, this manuscript is well-written, stating that the paper is interesting and correctly prepared.

Response 1: We appreciate the reviewer for the favorable analysis of our original submission and for highlighting the significance of our studies.

Specific comments:

Comment 1 for “Abstract”: In the abstract part, methods, it is essential to add the something about statistical analyses.

Response 1 for “Abstract”: We apologize for the inappropriate description here. statistical analyses was add to line20-21“All data were tested using paired samples T-test with a statistical significance level of P<0.05”

Comment 2 for “Abstract”: In the abstract part, results, main findings are needed to show and combine in this part.

Response 2 for “Abstract”: We apologize for the inappropriate description here. The results and mains findings have been showed and combine in the line 21-35

Comment 1 for “Introduction”: In the intro part, the first paragraph, “The tennis racket is a complex system composed of the frame, handle, and strings” It is recommended that add some relevant references here.

Response 1 for “Introduction”: “The tennis racket is a complex system composed of the frame, handle, and strings” was modified to “The tennis racket is composed of the frame, handle, and strings” . The relevant references is Analysis and experimental research on the mechanical characteristics of tennis rackets

Comment 2for “Introduction”: In the intro part, display the aim of this study rather than the necessary and hypothesis

Response 2for “Introduction”: The aim of the study was displayed in the line 71-74 of manuscript

Comment for “Methold”: In the methold part, the method is well-structured, with a clear and correct development of sections. It is better to add a figure of experimental set-up which includes participant, racket, net and other necessary equipment.

Response for “Method”:The figure of experimental set-up was added in Fig1. Fig2. Fig3 , Also the description of the “Experimental equipment”section has also been modified, The details see the line 85-102 of manuscript

Comment for “Results”: In the results part, tables included in the paper are clear, and there is extensive notes accompanying them. This suggests that the visual representations are effective in conveying information. However, the content of the results text is too short, try to add some details in order this section entirely.

Response for “Results”:The details have been added in the line 144-158,165-176,183-189

Comment for “Discussion”: In the discussion part, the authors should display the limitations of their study, which is appropriately discussed in the end.

Response for “Discussion”: We apologize for the inappropriate description here. The Limits of the study have been displayed in line 301-310, and discussed appropriately

---

## [Decision Letter · Decision Letter 1]

4 Jun 2024

PONE-D-24-00255R1Effects of different tennis racket string tension on forehand stroke effect and racket dynamic impactPLOS ONE

Dear Dr. Liu,

Thank you for submitting your manuscript to PLOS ONE. After careful consideration, we feel that it has merit but does not fully meet PLOS ONE’s publication criteria as it currently stands. Therefore, we invite you to submit a revised version of the manuscript that addresses the points raised during the review process.

We look forward to receiving your revised manuscript.

Kind regards,

Yaodong Gu

Academic Editor

PLOS ONE

Additional Editor Comments:

Better to put the main finding in the discussion.

Reviewers' comments:

Reviewer's Responses to Questions

**Comments to the Author**

1. If the authors have adequately addressed your comments raised in a previous round of review and you feel that this manuscript is now acceptable for publication, you may indicate that here to bypass the “Comments to the Author” section, enter your conflict of interest statement in the “Confidential to Editor” section, and submit your "Accept" recommendation.

Reviewer #2: All comments have been addressed

Reviewer #3: (No Response)

2. Is the manuscript technically sound, and do the data support the conclusions?

Reviewer #2: Yes

Reviewer #3: Partly

3. Has the statistical analysis been performed appropriately and rigorously? 

Reviewer #2: Yes

Reviewer #3: No

4. Have the authors made all data underlying the findings in their manuscript fully available?

Reviewer #2: Yes

Reviewer #3: Yes

5. Is the manuscript presented in an intelligible fashion and written in standard English?

Reviewer #2: No

Reviewer #3: No

6. Review Comments to the Author

Reviewer #2: The revised manuscript is well-written and it is able to be considered to publish. However, some issues have to be modified before publication.

1.The format of cited references in the introduction, such as Zhang W, Jean Bernard Fabre and others, needs to be revised.

2.Do not use "meanwhile" in the research article, it is a colloquial expression.

3.If humans were involved in the study, it is recommended use "participant" not "subject" in the method.

4.All the figures are not only blurry but chaotic background. It is better to change or replace figures.

5.It is feasible to combine the "Experimental equipment" with "Testing method" which could be called "Experimental Procedures".

6.The writing is of poor quality, please proof-read to improve the clarity for reading and understanding.

Reviewer #3: Review comment

This manuscript entitled “Effects of different tennis racket string tension on forehand stroke effect and racket dynamic impact” is encouraging research, but there are several questions that should be addressed before this manuscript can be accepted for publication, which are listed below. You can revise this paper more properly.

Specific comments

1. I suggest optimizing the structure of this manuscript, the format and framework of this manuscript need to be optimized.

2. The format of the abstract part does not meet the specifications and it is recommended to modify it.

3. The citation format of the reference does not comply with the standards of academic articles. It is recommended to modify it.

4. The introduction section lacks the hypothesis of this study. It is recommended to state the problem of this study and add the research hypothesis.

5. The meaning and application of this research should be further and clearly expressed in the introduction section. It will help readers to understand the function and value of this study. Please further explain why have to do this research and the possible benefit of this research to this field.

6. Demographic information on subjects is missing, which is necessary material.

7. In the method section, comprehensive information about the experimental equipment needs to be added in the method section, such as brand, model, company, etc.

8. The title of the figure needs to be placed below the figure.

9. In the Methods section, please add more details and descriptions to complete this part.

10. The experiment is missing a lot of detailed descriptions and process descriptions, which violates the principle of repeatability of the experiment. It is recommended that the description of the method section be re-edited.

11. I suggest adding the key findings at the beginning of the discussion section.

12. The application of key findings in this study should be further mentioned and discussed.

13. The language level of this manuscript needs improvement, and I recommend that the manuscript undergo a thorough grammatical and formatting check.

7. PLOS authors have the option to publish the peer review history of their article (what does this mean?). If published, this will include your full peer review and any attached files.

Reviewer #2: No

Reviewer #3: No

---

## [Author Response · Author response to Decision Letter 1]

29 Jul 2024

Specific comments:

Comment 1: The format of cited references in the introduction, such as Zhang W, Jean Bernard Fabre and others, needs to be revised.

Response 1: The format of cited references in the introduction has been revised in the line 59-74

Comment 2: Do not use "meanwhile" in the research article, it is a colloquial expression

Response 2: I’m sorry to this error, the mistake has been revised.

Comment 3: If humans were involved in the study, it is recommended use "participant" not "subject" in the method.

Response 3: Your profound language skills are worthy of my learning, and I have made modifications in the line 90 

Comment 4: All the figures are not only blurry but chaotic background. It is better to change or replace figures.

Response4: According to your suggestion, we have corrected the figures 

Comment5: It is feasible to combine the "Experimental equipment" with "Testing method" which could be called "Experimental Procedures".

Response 5: Thank you for your valuable advice, the "Experimental equipment" with "Testing method" have been calles "Experimental Procedures" in line 102-135

Comment 6: The writing is of poor quality, please proof-read to improve the clarity for reading and understanding.

Response 6: I’m sorry for that, we have improved the writing quality

Reviewer #3: This manuscript entitled “Effects of different tennis racket string tension on forehand stroke effect and racket dynamic impact” is encouraging research, but there are several questions that should be addressed before this manuscript can be accepted for publication, which are listed below. You can revise this paper more properly.

Specific comments:

Comment 1: I suggest optimizing the structure of this manuscript, the format and framework of this manuscript need to be optimized.

Response 1: Thank you for your valuable advice, we have optimizing the structure of this manuscript in the "Abstract","Methods"," Discussion"," References"

Comment 2: The format of the abstract part does not meet the specifications and it is recommended to modify it.

Response 2: I’m sorry for the mistake, the format of the abstract part has been modified in line 37-47, also check the format of the manuscript.

Comment 3: The citation format of the reference does not comply with the standards of academic articles. It is recommended to modify it.

Response3: PLOS uses the reference style outlined by the International Committee of Medical Journal Editors (ICMJE), also referred to as the “Vancouver” style, according to the “Vancouver” style, we modified the citation format of the reference in the line 344-448

Comment 4: The introduction section lacks the hypothesis of this study. It is recommended to state the problem of this study and add the research hypothesis.

Response 4: Thank you for you advise, we added the research hypothesis in the line84-88

Comment 5: The meaning and application of this research should be further and clearly expressed in the introduction section. It will help readers to understand the function and value of this study. Please further explain why have to do this research and the possible benefit of this research to this field.

Response 5: Thank you for your comment, we added the meaning and application of this research in line 85-88, we have explained why have to do this research in line 56-79

Comment 6: Demographic information on subjects is missing, which is necessary material.

Response 6:We have already added demographic information in line 91-101

Comment 7: In the method section, comprehensive information about the experimental equipment needs to be added in the method section, such as brand, model, company, etc.

Response 7: Than you for you comment, we added the comprehensive about the experimental equipment in line 117-128

Comment 8: The title of the figure needs to be placed below the figure.

Response 8: I’m sorry for that, PLOS Manuscript Body Formatting Guidelines in line 32-34 demand that “Each figure caption should appear directly after the paragraph in which they are first cited.” “Figures should be uploaded separately as individual files.” So the figure title and caption can appear in the Manuscript.

Comment 9: In the Methods section, please add more details and descriptions to complete this part.

Response 9: You comment is very important, we have written the Methods in line 91-134

Comment 10: The experiment is missing a lot of detailed descriptions and process descriptions, which violates the principle of repeatability of the experiment. It is recommended that the description of the method section be re-edited.

Response 10: Thank you for your advice, the detailed descriptions and process descriptions have been re-edited in line 103-152

Comment 11: I suggest adding the key findings at the beginning of the discussion section.

Response 11: Thank you for your valuable advice, the key findings has been added at the beginning of the discussion section in line 205-211

Comment 12: The application of key findings in this study should be further mentioned and discussed.

Response 12: The application of key findings in this study should be further mentioned and discussed in line 307-321

Comment 13: The language level of this manuscript needs improvement, and I recommend that the manuscript undergo a thorough grammatical and formatting check.

Response 13: I’m sorry for may poor language of this manuscript, In accordance with your suggestions and the professional polishing team, we have made changes to the grammar and format of the manuscript

---

## [Decision Letter · Decision Letter 2]

12 Aug 2024

PONE-D-24-00255R2Effects of different tennis racket string tension on forehand stroke effect and racket dynamic impactPLOS ONE

Dear Dr. Liu,

Thank you for submitting your manuscript to PLOS ONE. After careful consideration, we feel that it has merit but does not fully meet PLOS ONE’s publication criteria as it currently stands. Therefore, we invite you to submit a revised version of the manuscript that addresses the points raised during the review process.

We look forward to receiving your revised manuscript.

Kind regards,

Yaodong Gu

Academic Editor

PLOS ONE

Additional Editor Comments:

The authors shall check the format of the paper in accordance with journal's requirement.

Reviewers' comments:

Reviewer's Responses to Questions

**Comments to the Author**

1. If the authors have adequately addressed your comments raised in a previous round of review and you feel that this manuscript is now acceptable for publication, you may indicate that here to bypass the “Comments to the Author” section, enter your conflict of interest statement in the “Confidential to Editor” section, and submit your "Accept" recommendation.

Reviewer #2: All comments have been addressed

Reviewer #3: (No Response)

2. Is the manuscript technically sound, and do the data support the conclusions?

Reviewer #2: Yes

Reviewer #3: Partly

3. Has the statistical analysis been performed appropriately and rigorously? 

Reviewer #2: Yes

Reviewer #3: Yes

4. Have the authors made all data underlying the findings in their manuscript fully available?

Reviewer #2: Yes

Reviewer #3: Yes

5. Is the manuscript presented in an intelligible fashion and written in standard English?

Reviewer #2: Yes

Reviewer #3: No

6. Review Comments to the Author

Reviewer #2: • The introduction effectively presents the motivation and provides basic information, referring to other publications in the field.

• The publication is well-structured, with a clear and correct development of sections, including abstract, introduction, materials and methods, results, discussion, and conclusions.

• The graphs included in the paper are clear. This suggests that the visual representations are effective in conveying information.

• The authors demonstrate awareness of the limitations of their study, and this is appropriately discussed in the discussion section.

• The overall conclusion is positive, stating that the paper is interesting and correctly prepared. The only suggested improvement is related to the volume of the abstract, which is too long.

• After decreasing the volume of abstract, the paper is suitable for publication.

Reviewer #3: Review comment

Thanks to the author for the revision of the manuscript. There are some comments as

follows:

1. Line 40-42. In my point of view, this conclusion is unconvincing and not seriously.

The ‘stroke performance’ in racket sports usually focuses on the stroke play motions

and the performance of the athletes. However, this study only focuses on the impact

of the racket on the ball’s movement, so the use of "stroke performance" is not

rigorous.

2. In the Line 98-99. ‘.... using a forehand stroke directed cross-court, Each

participant…’ I am confusing about this section, maybe you need change ‘Each’ to

‘each’? Please check this issue of whole article.

3. In the Experimental Procedures section, the details are not enough to repeat this

study. Please mentioned is there a ball machine or a coach to do the serve? If it is a

machine or coach, how do you control the ball’s condition? Such as ball’s speed, fall,

and rotation? Please, add more necessary details.

4. Line 116-127. Please check the format and font of this section. I noticed lots of font

and formatting errors.

5. Line 167-170, There is a space in the line here, but there is no space between

L184-L186, why?

6. Line 107. Please revise the citation format.

7. In the Table 2, 3, and 4, there are some error and formatting errors, please check it.

8. Please check the format and font of whole article again. Because I noticed there are

lots of font formatting errors in the manuscript.

7. PLOS authors have the option to publish the peer review history of their article (what does this mean?). If published, this will include your full peer review and any attached files.

Reviewer #2: No

Reviewer #3: **Yes: **Yuqi He

---

## [Author Response · Author response to Decision Letter 2]

30 Aug 2024

Responds to the reviewers’ comments:

Reviewer #2: • The introduction effectively presents the motivation and provides basic information, referring to other publications in the field.

• The publication is well-structured, with a clear and correct development of sections, including abstract, introduction, materials and methods, results, discussion, and conclusions.

• The graphs included in the paper are clear. This suggests that the visual representations are effective in conveying information.

• The authors demonstrate awareness of the limitations of their study, and this is appropriately discussed in the discussion section.

• The overall conclusion is positive, stating that the paper is interesting and correctly prepared. The only suggested improvement is related to the volume of the abstract, which is too long.

• After decreasing the volume of abstract, the paper is suitable for publication.

Response: Thank you for your comments, we decreased the volume of abstract in line 14-39

Reviewer #3: Thanks to the author for the revision of the manuscript. There are some comments as follows:

Specific comments:

Comment 1: Line 40-42. In my point of view, this conclusion is unconvincing and not seriously. The ‘stroke performance’ in racket sports usually focuses on the stroke play motions and the performance of the athletes. However, this study only focuses on the impact of the racket on the ball’s movement, so the use of "stroke performance" is not rigorous.

Response 1: Thank you for your advice, according your view, “stroke performance” was change to “stroke effect” in line 31-32.

Comment 2: In the Line 98-99. ‘.... using a forehand stroke directed cross-court, Each participant…’ I am confusing about this section, maybe you need change ‘Each’ to‘each’? Please check this issue of whole article.

Response 2: I’m sorry for the mistake, we modified this in line 88-89, and check the issue of whole article.

Comment 3: In the Experimental Procedures section, the details are not enough to repeat this study. Please mentioned is there a ball machine or a coach to do the serve? If it is a machine or coach, how do you control the ball’s condition? Such as ball’s speed, fall, and rotation? Please, add more necessary details.

Response3: Before the test, we Before the formal test, we conducted a pre-experiment, adjusted the self-made ball feeding device to ensure the quality of the ball. The self-made ball feeding device is shown in the figure 2, the details description is shown in line 111-117

Comment 4: Line 116-127. Please check the format and font of this section. I noticed lots of font and formatting errors.

Response 4: Thank you for your comment, we revised the format and font of this section in the line104-119. The figure captions in line 111,114,118 use bold type for the figure titles according the manuscript body formatting guidelines.

Comment 5: Line 167-170, There is a space in the line here, but there is no space between L184-L186, why?

Response 5: I’m sorry for the stupid mistake, we correct the mistake and check the hole manuscript

Comment 6: Line 107. Please revise the citation format.

Response 6: Thank you, We have already revised the citation format in line 95-96

Comment 7: In the Table 2, 3, and 4, there are some error and formatting errors, please check it.

Response 7: Thank you for the comment, we revised the error in the Table 2,3,4.

Comment 8: Please check the format and font of whole article again. Because I noticed there are lots of font formatting errors in the manuscript.

Response 8: Thank you for this comment, We check the full manuscript and modified the errors.

---

## [Decision Letter · Decision Letter 3]

24 Sep 2024

PONE-D-24-00255R3Effects of different tennis racket string tension on forehand stroke effect and racket dynamic impactPLOS ONE

Dear Dr. Liu,

Thank you for submitting your manuscript to PLOS ONE. After careful consideration, we feel that it has merit but does not fully meet PLOS ONE’s publication criteria as it currently stands. Therefore, we invite you to submit a revised version of the manuscript that addresses the points raised during the review process.

We look forward to receiving your revised manuscript.

Kind regards,

Yaodong Gu

Academic Editor

PLOS ONE

Journal Requirements:

**Additional Editor Comments:**

Please check some minor questions raised by the reviewer.

Reviewers' comments:

Reviewer's Responses to Questions

**Comments to the Author**

1. If the authors have adequately addressed your comments raised in a previous round of review and you feel that this manuscript is now acceptable for publication, you may indicate that here to bypass the “Comments to the Author” section, enter your conflict of interest statement in the “Confidential to Editor” section, and submit your "Accept" recommendation.

Reviewer #2: All comments have been addressed

Reviewer #3: All comments have been addressed

2. Is the manuscript technically sound, and do the data support the conclusions?

Reviewer #2: Yes

Reviewer #3: Partly

3. Has the statistical analysis been performed appropriately and rigorously? 

Reviewer #2: Yes

Reviewer #3: Yes

4. Have the authors made all data underlying the findings in their manuscript fully available?

Reviewer #2: Yes

Reviewer #3: Yes

5. Is the manuscript presented in an intelligible fashion and written in standard English?

Reviewer #2: Yes

Reviewer #3: Yes

6. Review Comments to the Author

Reviewer #2: After previous issues solved, the present revised manuscript is well-written, and it is able to be considered for publication.

Reviewer #3: There are still some issue about format should be revised. Please check the whole manuscript again, thanks.

7. PLOS authors have the option to publish the peer review history of their article (what does this mean?). If published, this will include your full peer review and any attached files.

Reviewer #2: No

Reviewer #3: **Yes: **Yuqi He

---

## [Author Response · Author response to Decision Letter 3]

22 Oct 2024

Responds to the reviewers’ comments:

 Comment 1: 

Is the manuscript technically sound, and do the data support the conclusions? The manuscript must describe a technically sound piece of scientific research with data that supports the conclusions. Experiments must have been conducted rigorously, with appropriate controls, replication, and sample sizes. must be drawn appropriately based on the data presented.

Reviewer #3: Partly

Response：We are sorry for our mistakes. After examining our manuscript, we found there may be some unclear and insufficient descriptions of the results and conclusions, especially in the Abstract section. 

Therefore, we enriched our results, and rewrote the conclusions to make them correspond to our results in lines 35-42.

Comment 2: 

Reviewer #3: There are still some issues about format should be revised. Please check the whole manuscript again, thanks.

Response：Sorry for the issues about format, and we have made some modifications in the manuscript, such as below:

1) In line 114: the brand of tennis balls used in the test was described more precision and clarity.

2) In Table 4 and lines 190-191: we used more standardized and accurate statistical significance annotation symbols.

3) In lines 322-328: we rewrote the conclusions to make them correspond to our results.

---

## [Decision Letter · Decision Letter 4]

13 Nov 2024

PONE-D-24-00255R4Effects of different tennis racket string tension on forehand stroke effect and racket dynamic impactPLOS ONE

Dear Dr. Liu,

Thank you for submitting your manuscript to PLOS ONE. After careful consideration, we feel that it has merit but does not fully meet PLOS ONE’s publication criteria as it currently stands. Therefore, we invite you to submit a revised version of the manuscript that addresses the points raised during the review process.

We look forward to receiving your revised manuscript.

Kind regards,

Yaodong Gu

Academic Editor

PLOS ONE

Journal Requirements:

Reviewers' comments:

Reviewer's Responses to Questions

**Comments to the Author**

1. If the authors have adequately addressed your comments raised in a previous round of review and you feel that this manuscript is now acceptable for publication, you may indicate that here to bypass the “Comments to the Author” section, enter your conflict of interest statement in the “Confidential to Editor” section, and submit your "Accept" recommendation.

Reviewer #3: All comments have been addressed

2. Is the manuscript technically sound, and do the data support the conclusions?

Reviewer #3: Yes

3. Has the statistical analysis been performed appropriately and rigorously? 

Reviewer #3: Yes

4. Have the authors made all data underlying the findings in their manuscript fully available?

Reviewer #3: Yes

5. Is the manuscript presented in an intelligible fashion and written in standard English?

Reviewer #3: Yes

6. Review Comments to the Author

Reviewer #3: Thank you for all the edits and adjustments you made to this manuscript. There are still have some issue should be adjust.

7. PLOS authors have the option to publish the peer review history of their article (what does this mean?). If published, this will include your full peer review and any attached files.

Reviewer #3: No

---

## [Author Response · Author response to Decision Letter 4]

27 Dec 2024

Specific comments

 Comment 1: 

Reviewer #3: 1. The manuscript mentions that the string tension of a tennis racket has an important influence on playing performance and material properties, but does not elaborate on why string tension is an important area of research. Some background information could be added, such as how string tension affects the performance of professional and amateur players and its importance in the selection of tennis equipment.

Response：Thank you for your comment, we have added the background information in line 58-60.

Comment 2: 

Reviewer #3: 2. Please be more specific about why you chose forehand strokes as the subject of your study and why you chose the International Tennis Number (ITN) as the evaluation criterion.

Response：We are sorry for the stupid mistakes, we have explained why you chose forehand strokes as the subject of our study in line 44-53, and Modifications have been made regarding why ITN was chosen as the evaluation standard in line 69-75.

Comment 3: 

Reviewer #3: 3. L58-L59, ‘Much of the prior research was conducted under laboratory conditions, lacking objective standards for evaluating stroke control based on participant performance’ …The journal encourages the addition of articles with more recent years. The following literature is encouraged to be cited and referenced: ‘He Y, Fekete G, Sun D, Baker JS, Shao S, Gu Y. Lower Limb Biomechanics during the Topspin Forehand in Table Tennis: A Systemic Review. Bioengineering. 2022; 9(8):336. https://doi.org/10.3390/bioengineering9080336’.

Response：We have add the articles published in recent years, including the article you recommended.

---

## [Editor Report · Decision Letter 5]

30 Dec 2024

Effects of different tennis racket string tension on forehand stroke effect and racket dynamic impact

PONE-D-24-00255R5

Dear Dr. Liu,

We’re pleased to inform you that your manuscript has been judged scientifically suitable for publication and will be formally accepted for publication once it meets all outstanding technical requirements.

Kind regards,

Yaodong Gu

Academic Editor

PLOS ONE
---

## [Editor Report · Acceptance letter]

3 Jan 2025

PONE-D-24-00255R5 

PLOS ONE

Dear Dr. Liu, 

I'm pleased to inform you that your manuscript has been deemed suitable for publication in PLOS ONE. Congratulations! Your manuscript is now being handed over to our production team.

Kind regards, 

on behalf of

Professor Yaodong Gu 

Academic Editor

PLOS ONE